# IvoryOS: an interoperable web interface for orchestrating Python-based self-driving laboratories

Wenyu Zhang [1] ✉, Lucy Hao[1], Veronica Lai[2], Ryan Corkery[2], Jacob Jessiman[1], Jiayu Zhang[1], Junliang Liu[2], Yusuke Sato[2], Maria Politi[1], Matthew E. Reish[1], Rebekah Greenwood[1], Noah Depner[1], Jiyoon Min [1], Rama El-khawaldeh[1], Paloma Prieto [1,2], Ekaterina Trushina [1] & Jason E. Hein [1,2,3] ✉

Self-driving laboratories (SDLs), powered by robotics, automation and artificial intelligence, accelerate scientific discoveries through autonomous experimentation. However, their adoption and transferability are limited by the lack of standardized software across diverse SDLs. In this work, we introduce IvoryOS – an open-source orchestrator that automatically generates web interfaces for Python-based SDLs. It ensures interoperability by dynamically updating the user interfaces with the plugged components and their functionalities. The interfaces enable users to directly control SDLs and design workflows through a drag-and-drop user interface. Additionally, the workflow manager provides no-code configuration for iterative execution, supporting both human-in-the-loop and closed-loop experimentation. We demonstrate the integration of IvoryOS with six SDLs across two institutes, showcasing its adaptability and utility across platforms at various development stages. The plug-and-play and low-code feature of IvoryOS addresses the rapidly evolving demands of SDL development and significantly lowers the barrier to entry for building and managing SDLs.

Self-driving laboratories (SDLs) integrate commercial or custom-designed hardware, such as liquid handlers, automated reactors, analytical instruments, robotic arms, etc., to automate chemistry experimentation[1–3]. Utilizing machine learning algorithms, such as Bayesian Optimization, SDLs autonomously suggest subsequent experimental conditions based on prior data to close the loop in scientific discovery[1,4,5]. The SDL prototypes emerge in various research domains, including materials chemistry, drug discovery, and formulation[6–10], with diversity in hardware components and overall form factors. Most researchers often need to develop custom scripts to orchestrate hardware, data pipeline and experimental planner modules. Python is dominating for SDL orchestration layer given its open-source and available libraries in hardware communication[11–13], data analysis and machine learning[14–16]. Despite Python's capability and

open-source contributions from the lab automation community[17–22], it still poses significant challenges to build, maintain and operate an SDL without an adequate programming background.

Graphical User Interfaces (GUIs) can enhance the accessibility of SDLs, but typically require substantial expertise and time to develop and are often hardware-specific. Accounting that most SDLs use multiple hardware and prioritize flexible automation for efficient reconfiguration[23,24], it further reframes GUI development towards general-purpose solutions. Previous work, such as ChemIDE, enables visual programming using chemical description language (χDL), a hardware-independent language that allows chemists to script transferable and machine-readable chemical syntheses[25–27]. SDL developers can map their own hardware control scripts to χDL operations, enabling the execution of standardized χDL scripts across various

[1]Department of Chemistry, The University of British Columbia, Vancouver, BC, Canada. [2]Telescope Innovations Corp, Vancouver, BC, Canada. [3]Department of Chemistry, University of Bergen, Bergen, Norway. ✉e-mail: ivoryzhang@chem.ubc.ca; jhein@chem.ubc.ca

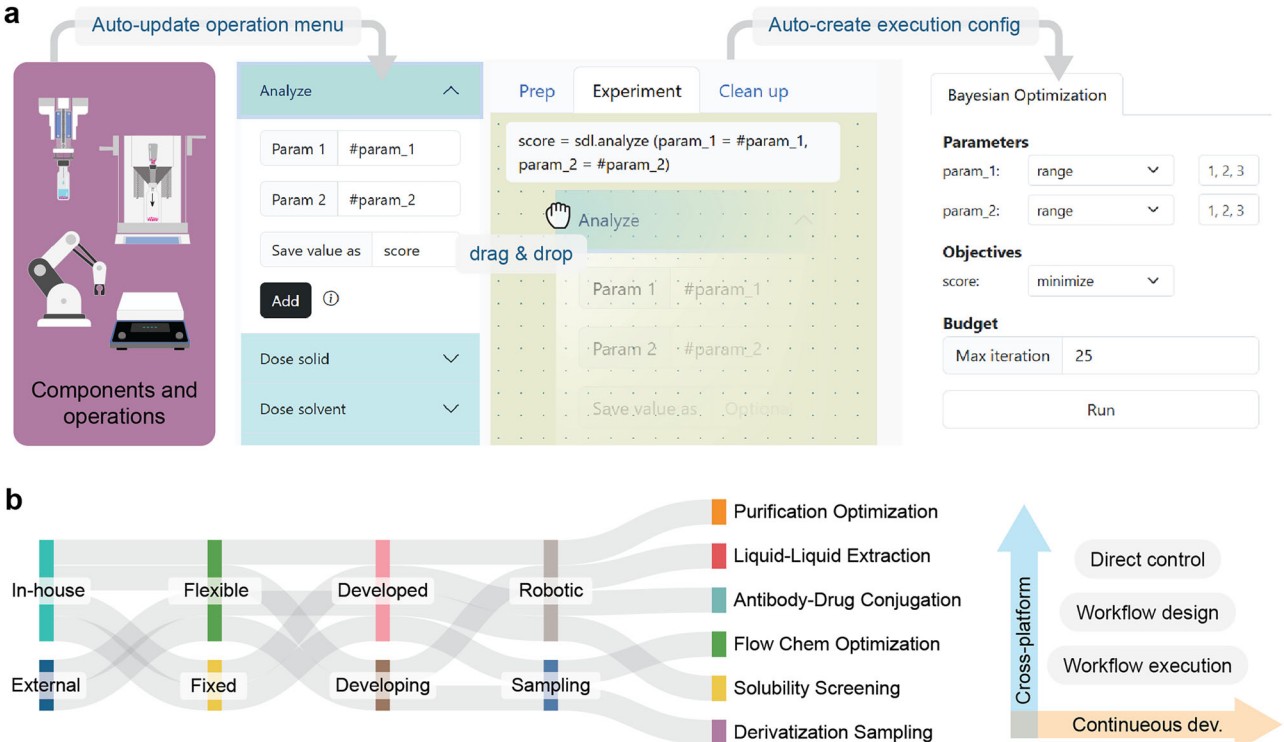

**Fig. 1 | IvoryOS software feature and its interoperability across six self-driving laboratories (SDLs). a** Illustration of plug-and-play architecture with screenshots of workflow design and configuration interfaces. When launching IvoryOS within the Python script that includes component instances, it automatically updates the operation menu in the web interfaces by serializing the initialized modules in the script. The featured workflow design interface allows low-code workflow design and can create a closed-loop configuration for scripts with dynamic parameters. **b** IvoryOS was integrated with six SDLs, both in-house and external, demonstrating its cross-platform interoperability. For platforms that evolve for various research objectives, the plug-and-play architecture also supports continuous development.

systems[28,29]. AlabOS focused on developing a reconfigurable workflow status management framework along with the dedicated GUI for visualization[30,31]. Developers can tailor their workflow scripts into the AlabOS framework, thereby enabling managing complex SDL operations within an established GUI. ChemOS 2.0 is an orchestration platform, focusing on experimental planning with real-time execution in the chemistry domain[32]. It supports specific modular device communication through SiLA2 with simulation mode, making it adaptable across all stages of SDL development. Despite these advancements, quick adaptation of these GUIs to existing workflows is not feasible due to the workload and expertise needed for framework tailoring (AlabOS and ChemOS 2.0) and local runtime configuration (χDL).

In this work, we propose IvoryOS, an open-source, interoperable web interface for Python-based SDLs. This system provides a configuration-free solution for operating diverse SDLs and accommodating continuously developing requirements. Specifically, a platform state capture is generated each time to ensure up-to-date modules, functionalities, and input fields in the operation menu (Fig. 1a). The web interfaces include direct control, workflow design, and workflow execution management, addressing the essential needs when operating an SDL. We demonstrate the adaptability of IvoryOS on six SDLs at two laboratories with various hardware components, development stages, and research objectives (Fig. 1b). We also showcase the ease of configuring closed-loop experimentation with two case studies.

## Results

### Web server architecture

In this robot control architecture using Flask, a lightweight web application framework for Python, the source SDL script is dynamically loaded into the system upon launching the server, allowing the

backend to access the methods (Fig. 2). Within the backend, a form module dynamically generates control and design forms for all serialized methods, which can be rendered in web user interfaces. Operators can interact with the SDL through direct function calling control or workflow design through Hypertext Transfer Protocol (HTTP). A workflow script module handles editing, storing, and workflow validation, allowing either online or offline access of workflow scripting using SDL abstractions. The script control module, interfacing with connected hardware systems, then executes scripted and compiled functions according to the iteration plan. Execution status and real-time feedback are communicated to the interface via WebSocket (Fig. 2).

### Direct control interface

The displayed web forms within the direct control interface show all available methods of an SDL or laboratory equipment. Figure 3a shows a mapping between the control interface and the source code methods. The method name, parameter names, type hints, and default values are parsed to the input forms in the control interface. Users can navigate through available methods and interact with the SDL by inputting required parameters. The control interface can be customized and reorganized to suit frequent usage and visibility preferences. Such flexibility enhances the usability and adaptability of the interface across various SDL frameworks or individual laboratory equipment Application Programming Interfaces (APIs)[19–21], permitting a plug-and-play GUI for prototyping hardware prior to the development of a dedicated GUI.

### Workflow design interface

Figure 3c shows the component modules that are serialized from the Python source code. A flow control component, including

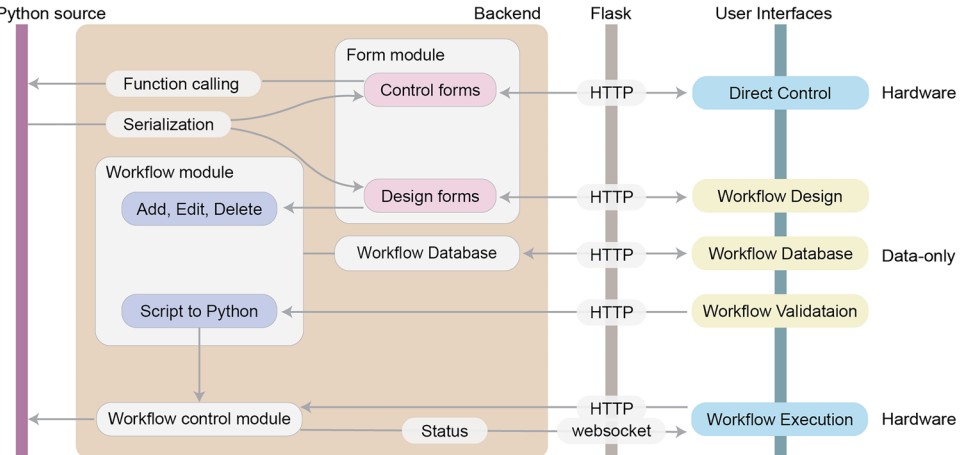

**Fig. 2 | IvoryOS architecture and interoperable SDL control through script serialization.** The backend automatically creates an SDL abstraction and interactive web forms based on the user interface types (direct control or workflow design). The direct control interface renders control forms for handling parameter input, while the backend executes the task through function calling. The workflow design interface renders design forms and allows low-code programming using the operations in the source Python code. The workflow script is organized as a dictionary by the workflow script module for visually adding, editing, and deleting action steps. A database using the script model schema handles storing and managing workflow scripts. The workflow validation handles compilation of the workflow script back to an executable Python script. These requests in yellow are data-only workflow interactions and can be accessed offline without a hardware connection. On the other hand, the workflow execution requires the physical hardware connection, where the script control module handles the workflow execution request after a successful script validation. The script control module communicates with the web interface in real-time by emitting execution status via WebSocket.

fundamental logic constructs (if, while, and repeat), variable definition, and time delays (wait), enables the design of recursive or conditional workflows. To facilitate experimental design through visual programming, the parameter input fields are enhanced with a save field for storing function return values (Fig. 3d). Operators can drag the method card to the canvas area to configure the method input or click the `Add` button upon entering function parameters. The workflow is scripted into a dictionary collection with the workflow phases as keys for a list of step descriptions. The workflow script is visually represented on the design canvas, where users can interactively reorder steps or adjust task parameters. The design can also be exported as JavaScript Object Notation (JSON) format (Supplementary Information, Section 1). Once the design is complete, users can click the `Compile and Run` button to validate their design and convert the script to Python functions that are executable in the workflow execution page.

To enhance experimental design intuitiveness, each operation module in Fig. 3c optionally integrates a text-to-code card that attempts to translate natural language task descriptions into workflow script (Fig. 3e). Our previous work successfully translated literature into machine-executable scripts using large language models like GPT-4[33]. The translation is achieved through strategic prompting by giving sample code and instructions that map chemistry task descriptions to machine operations. In the context of flexible automation, where device functionalities are highly adaptable and lack strict operation mapping rules, we adapt this approach by appending the SDL code abstraction instead of defined mapping rules. An example of the expected JSON format used for visual workflow programming is also attached to the prompt, ensuring successful display on the design canvas. Additionally, a post-processing step validates the scripted functions to further guarantee the alignment with the module capabilities. The full prompt using the example abstract SDL is in Supplementary Information, Section 2.

### Workflow phases and parameters

Each workflow consists of three experimental phases: a preparation phase for preliminary steps, like purging solvent lines; a main experimental phase involving processes such as mixing, heating, purification,

etc.; and a cleanup phase to reset the platform post-experiment. Users can switch between editing phases using the tabs on the top of the workflow canvas (Fig. 3d). While the preparation and cleanup phases execute once with only constant parameters, the experiment phase is repeatable and supports dynamic parameters (Fig. 4a). The use of configurable parameters, in contrast to a constant value, can change during execution, allowing different iterations without modifying the workflow. In the workflow design interface, users can define a configurable (dynamic) parameter by employing the "#" notation followed by the parameter name (Fig. 3d; Analyze method input fields). This feature allows for flexible iterations with manual parameter entries or adaptive parameter suggestions through an optimization algorithm (Fig. 4b, see Workflow execution section).

### Workflow execution

On the execution interface, the available execution options depend on the parameter type and the presence of output values (Fig. 4b). The repeat option is designed for workflows with constant parameters, allowing either single or repeated executions (Fig. 4c). For configurable iterations, parameter entries with typing hints are populated for human-in-the-loop experimental design (Fig. 4d). For more efficient input, users can also upload a CSV file filled with parameter values for each iteration. Additionally, for flexible workflows that yield at least one numerical output, the Bayesian Optimization option with the Ax platform[34] becomes available, enabling adaptive exploration of the experimental space to optimize results (Fig. 4e). In this interface, all defined parameters and output values will be listed in the Parameters and Objectives sections respectively (Supplementary Information, Section 3). Note that only numeric outputs can be used as optimization objectives; non-numeric outputs should have their optimization option set to "none".

### Cross-platform integration

Figure 5 shows six distinct SDLs across the Hein Group and Telescope Innovations Corp. (Telescope) that have successfully integrated the IvoryOS as their orchestrator. These platforms integrate key components such as robotic arms, liquid handlers, temperature control units, automated balances, and analytical techniques like High-Performance

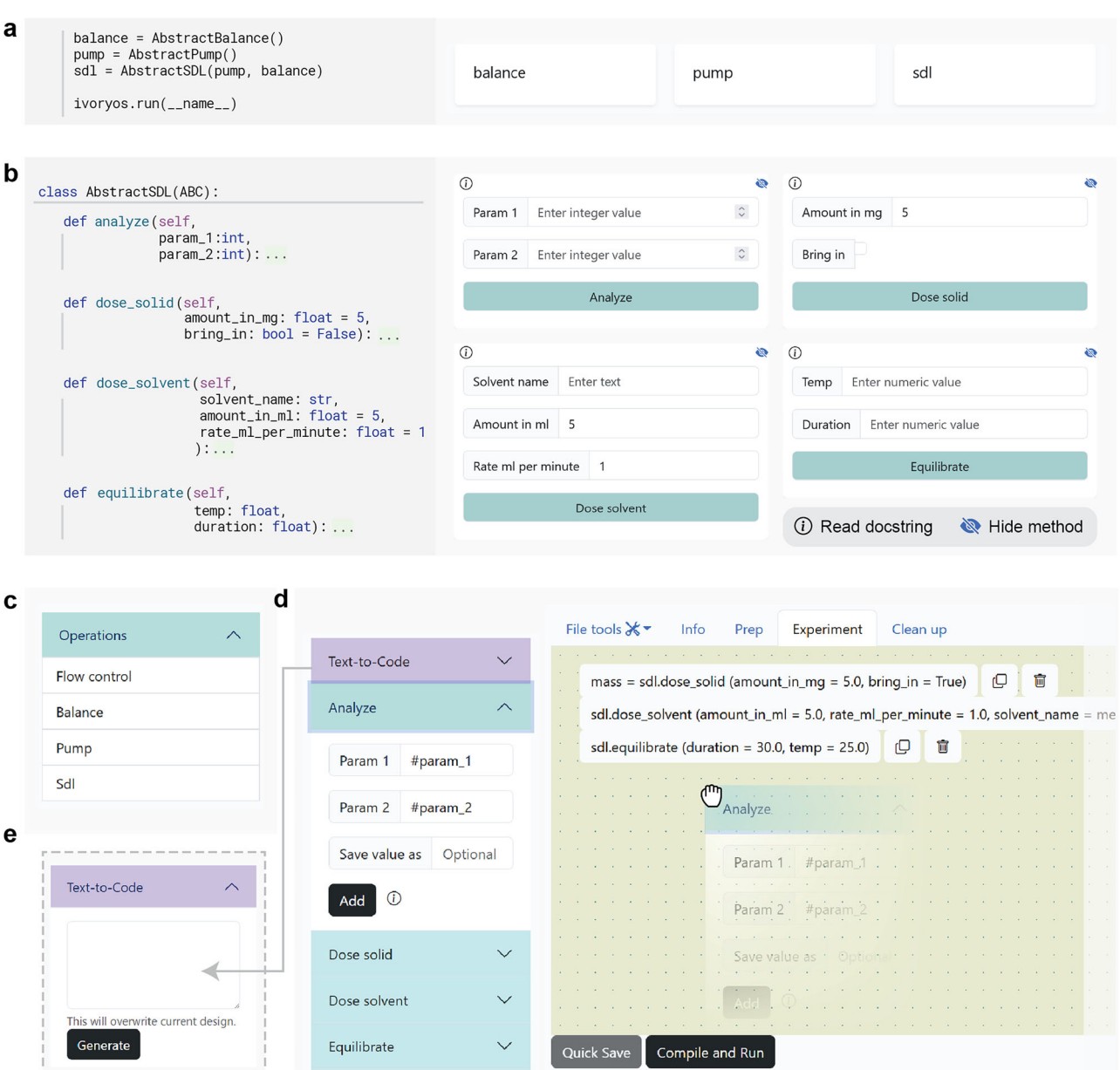

**Fig. 3 | IvoryOS direct control and workflow design interfaces using an abstract self-driving lab (SDL) script as an example. a** Automatic module recognition from custom instances in Python code. IvoryOS introspects the script to expose balance, pump, and sdl as accessible modules in the web interface. **b** Automatic web interface generation from Python class methods. Each method in the class is dynamically rendered as an interactive web form, with type hints and default values translated into appropriate input fields. **c** Captured device modules and a built-in flow control module that is available in the operation menu in workflow design interface. **d** Drag-and-drop workflow scripting that allows sequential task stacking and configurable parameter inputs. The #param_1 and #param_2 in the analyze method card are used to define configurable input for later execution. The top menu bar shows the current editing workflow phase and allows toggles between different phases. **e** Optional text-to-code feature for translating task description in natural language into workflow design using Large Language Model (LLM).

Liquid Chromatography (HPLC). While some systems focus on robotic arms for automating sample preparation and separation tasks (Fig. 5a, b, d, e), others focus on continuous sampling with DirectInject™ technology (Fig. 5c, f). The development approaches and hardware selections vary across platforms, reflecting various applications and research objectives. For example, although both are designed for solubility screening, the Telescope Solubility platform features a computer vision module instead of an HPLC in the in-house Purification platform (PurPOSE)[35]. For this reason, a filtration module is included in the PurPOSE platform but not in the Solubility platform for sample preparation. Similarly, the Antibody Drug Conjugation (ADC) platform focuses on automating small-volume liquid handling and sample washing for ADC, where a weighing module is incorporated instead of automated balances used in other robotic platforms that require powder handling[10,36]. The Liquid-Liquid Extraction (LLE) Platform is a platform under construction for scale-up automation, where an automated lab reactor is utilized for larger-scale and precisely-controlled agitating and tempering. The development stages, hardware components, and framework integrations are summarized in Supplementary Information Tables S1–S3.

Although the SDL development and frameworks exhibit variations across institutes and developers, Fig. 6 showcases a typical bottom-up SDL workstation development structure using the components from the PurPOSE platform (Fig. 5b). The development begins with

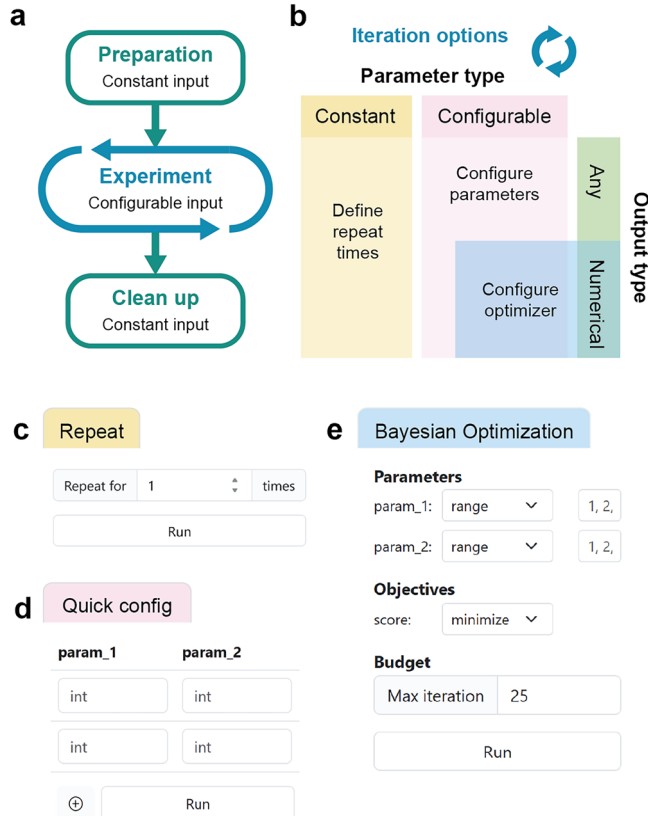

**Fig. 4 | Workflow phases and execution configuration interfaces based on parameter and output types. a** Workflow phases, including sequential execution of preparation, experiment, and clean up. The configuration allows diverse iterations for the experiment phase, while preparation and cleanup phases will only be executed once with constant inputs. **b** Execution options for the experiment phase based on the parameter and output types. Constant parameters allow only repeat time input. Configurable parameters allow configurable inputs or adaptive iteration through an optimization algorithm when there is a numerical output. **c** A screenshot of a fixed iteration interface. **d** A screenshot of configurable iteration interface using online form entry. **e** A screenshot of adaptive iteration interface with automatically generated parameters and objective configuration menu.

individual hardware components that usually support Python control with open-source APIs thanks to the SDL community[19–21]. The development process evolves into intermediate tasks that collaborate between robotic arm and other components to handle tasks, such as vessel transportation, vessel cap opening, and liquid transfer. The resulting user functions are usually capable of handling domain-specific tasks and are the building blocks for workflows. In addition to robotic workflow development, elements such as database, logging system, and scheduling frameworks can also be added for system robustness.

## Human-in-the-loop

Integration of the IvoryOS may offer diverse applications and flexibility when initializing at different layers of the development hierarchy. For example: (1) For platforms that have well-developed workflows, interfacing at the workflow level allows inputs for experimental conditions as well as iterative workflow configuration (high-throughput). The application of IvoryOS enhances usability by providing an intuitive interface for configuring parameters in pre-built workflows in the ADC and PurPOSE platforms (Fig. 5a, b). (2) Interfacing at the user function layer facilitates high-level task control and the creation of customized workflows for robotic arm-centric systems, such as PurPOSE (Fig. 5b)

and the Telescope Solubility platform (Fig. 5e). With the option of accessing pre-built experiments simultaneously, this capability supports task-specific adjustments, which are critical for experimental flexibility. (3) For ongoing platform development, interfacing with both intermediate tasks and hardware level can serve as a GUI for low-code robotic action builder, where users can either execute the workflow as-is or export the workflow in Python for future development. (4) Interfacing directly at the hardware component layer simplifies SDL development by providing facile and intuitive access to hardware APIs. In Fig. 5f, the derivatization sampling system at Telescope Innovations features an autosampler and the DirectInject™ system, assembled to improve the analysis of Lithium compounds in HPLC. IvoryOS served as a plug-and-play orchestrator, enabling no-code components coordination and workflow development using the module's API directly[19]. Supplementary Information S4 presents the platform's source code and workflows designed using IvoryOS.

## Closed-loop Experimentation

To highlight the autonomous experimental capabilities, we executed proof-of-concept closed-loop experiments on two platforms. On the Telescope Solubility platform (Fig. 5e), a computer vision-based color-matching optimization script was performed as a closed-loop experiment. The platform, controlled via IvoryOS, adjusted proportions of red and blue food coloring to achieve a predefined target color. With the Ax platform[34] serving as the brain of self-driving, the system iteratively fine-tuned mixing ratios according to the difference score analyzed through computer vision (Supplementary Information, Section S4.3.2). The execution demonstrates the no-code solution of enabling a data-driven optimization routine on a robotic platform that does not initially feature any optimizer configuration. Another closed-loop example was conducted on the Flow Chemistry platform (Fig. 5d) to optimize reaction conditions iteratively. The workflow controlled the flow rates and temperature based on the reaction conditions and monitored HPLC data for scoring the current condition. Despite that the platform was developed with an optimizer, the algorithm was limited to a discrete parameter space, thus limiting its usability and scalability. Deploying IvoryOS on the Flow Chemistry platform enables a more flexible and robust option for experimental design and permits seamless reconfiguration in the source code (Supplementary Information, Section S4.6).

## Discussion

We have demonstrated an adaptable approach to controlling SDLs through the integration of IvoryOS with six automated platforms. By dynamically generating a capture of Python scripts, it ensures that the GUI is not rigidly tied to predefined configurations. This feature allows users to rapidly adjust to new instruments, operations or processes without needing to rewrite or manually configure the orchestration layer. Such adaptability is particularly valuable for SDLs in chemistry, where standardized protocols are less established compared to automation in the life science sector. It often requires flexible functionalities and continuous platform development to accommodate dynamic research objectives[2,23,24]. Additionally, launching the IvoryOS server requires only a single line of code and does not necessitate any modifications to the Python script, further guaranteeing its adaptability to workflows at various stages of development. Unlike existing solutions, the quick adaptation is highlighted by a 10- to 30-min first-time installation and integration. The benefit of a web server architecture also supports remote access, facilitating collaboration with other analytical instruments or SDLs.

A key advantage of providing interoperable and configuration-free GUI is the democratization of SDL technology, particularly by lowering the barrier to entry for users unfamiliar with Python or programming in general. Traditionally, SDLs and other robotic control systems have required users to possess a certain level of coding

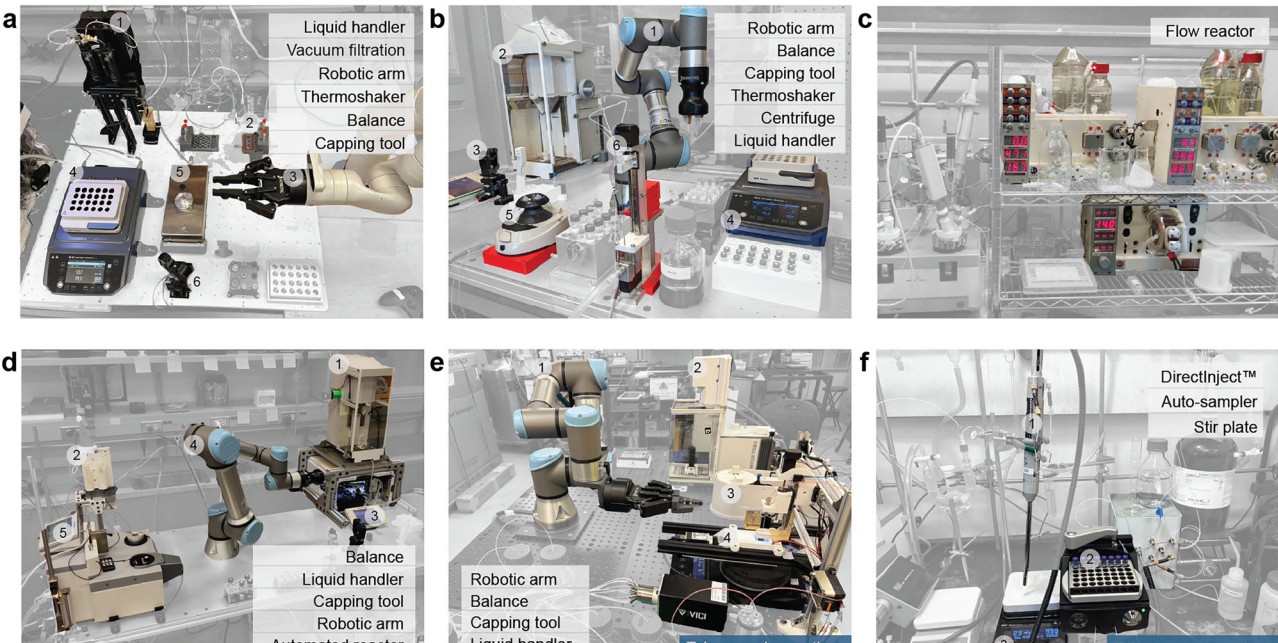

**Fig. 5 | Photographs and hardware components of various robotic arm-based or synthesis-focused Self-Driving Laboratories (SDLs) that have successfully integrated IvoryOS.** These platforms feature diverse hardware components and were developed for various research objectives. **a** The Antibody–Drug Conjugation (ADC) platform is a robotic arm-based platform for automated ADC synthesis and characterization[10]. **b** The PurPOSE Purification platform, developed as a multi-purpose platform for screening solubility and optimizing purification conditions[35].

**c** The Flow Chemistry platform, featuring Vapourtec flow reactors, is developed for reaction monitoring and optimization. **d** The Liquid-Liquid Extraction (LLE) Platform, currently under construction, is designed for automated scale-up LLE. **e** The Solubility platform at Telescope Innovations is a robotic platform that features computer vision-based solubility screening. **f** The derivatization sampling platform at Telescope Innovations is developed for automated online reaction monitoring with DirectInject™.

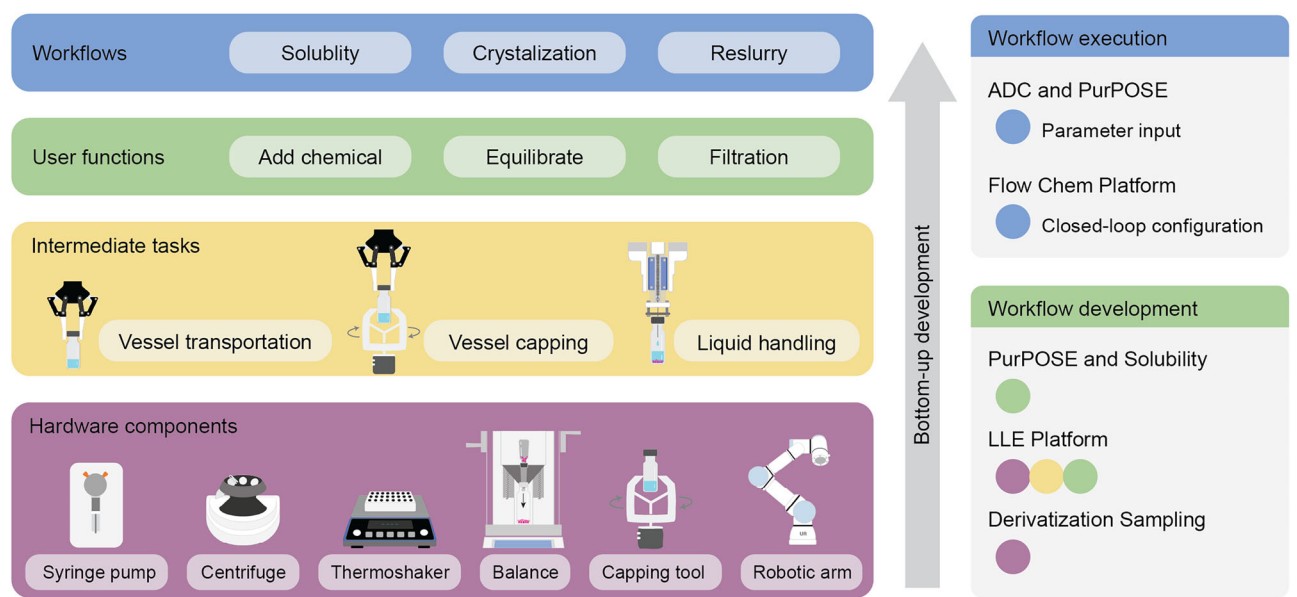

**Fig. 6 | Self-Driving Laboratories (SDLs) bottom-up development structure illustrated using the Hein Group PurPOSE Purification Platform components and functionalities; and color alignment to the integrated hierarchy across six SDLs.** The structure demonstrates the progression from individual hardware components to high-level user functions and composed experiments. Developed platforms with fixed automation typically expose only the workflow layer to users, while platforms with flexible automation may expose different layers to permit more flexible workflow design.

proficiency. However, with IvoryOS, users can now interact with the system through user-friendly interfaces without needing to code manually. This significantly lowers the learning curve, making advanced laboratory automation accessible to a broader scientific community. In the meantime, GUI development for human-robot interaction is a laborious task, especially with the need for autonomous decision-making. GUI development usually starts at the deployment stage, resulting in the lack of real user and trial-and-error during the developing stage. The use of an interoperable orchestrator permits anytime accessibility to the platform, streamlining the development

and testing cycle for SDLs. In this way, SDL platforms promote a more inclusive approach, empowering researchers across various scientific disciplines to leverage SDLs while focusing on solving domain-specific scientific problems.

While IvoryOS is primarily designed for SDLs and laboratory automation, the implementation of dynamic function capture and drag-and-drop GUI enables flexible orchestration of robotic or computational workflows. The no-code closed-loop execution may be applied to automate hyperparameter tuning in machine learning. This solution for achieving interoperability is equally valuable in building orchestration systems in other programming languages.

Despite the ease of adaptation, the software itself does not provide additional safety checking or workflow validation. It had been a challenge to address all possible safety precautions for flexible automation, compared to those of a fixed system. Thus, utilizing such interoperable software requires the developers to only expose safe tasks to the users and handle safety checks for individual tasks for potential sequential action stacking. Despite several recent advances in SDL scheduling frameworks[30,37], parallelization and scheduling remain a challenge for framework-free automation and interoperable software suite. This leads to the single-threaded nature when executing workflows using IvoryOS.

Looking ahead, we plan to enhance IvoryOS by introducing concurrent workflow designs and investigating compatibility with other scheduling software, aiming to allow parallel workflows that could significantly improve efficiency and flexibility. Improving the text-to-code robustness and accuracy is another direction to enhance the shareability of workflow between SDLs with similar functionality, where the workflow can be transferable through natural languages. Additionally, we aim to incorporate modular interfaces, enabling users to tailor the interface to their specific needs—whether focusing solely on control, omitting the database, or selecting other customized configurations.

## Methods

### Source code serialization

The serialization follows the Object-Oriented Programming (OOP) paradigm that is largely used for open-source lab instrument APIs and modular lab automation within the SDL communities. From the main SDL Python script, IvoryOS begins by systematically iterating over object variables in the current scope, finding instances that are of custom Python class types (Supplementary Information, Fig. S11). This ensures that only instances of user-defined methods relevant to the SDL functions are captured. Next, IvoryOS delves into each class's methods, selectively extracting those that are public and non-property functions. The retrieved parameter signatures and associated docstrings are then organized into a structured dictionary, with each entry corresponding to an SDL component. The resulting abstraction captures the operation modules, functionalities, and expected argument types of devices for further function calling or workflow designing. The serialization of the SDL is done by creating a .pkl file, facilitating offline access to the experimental design without the need for direct hardware connection. A complete abstraction result is in Supplementary Information, Section 5.

### Web form generation

For automatic updates in the operation menu, a web form is created for each method using WTForms Flask extension library. Each form is created with tailored input fields with type hints and default values according to the method signature (Fig. 3b). For primitive data types (int, float, str, and bool), the input fields natively support direct input and type validation. However, due to the limitation of text input fields, type validation is not supported for user-defined data types or collection types (list, tuple, dict, and set). This process enables a basic visualization of Python methods, allowing parameter input through a web interface. These forms are rendered in the direct control and workflow design interfaces (Fig. 3).

### Workflow database

The workflow database interface is implemented using Flask-SQLAlchemy to facilitate comprehensive management of experimental workflows within the system. The workflow script class extends the SQLAlchemy database model for data manipulation. This interface supports functionalities for saving, organizing, and loading workflows into the design interface from the database. Users can edit existing non-protected workflows, with changes being saved to the same or new workflows.

## Data availability

The IvoryOS source code, examples of six SDL integrations along with tutorial video using an abstract SDL have been deposited in https://gitlab.com/heingroup/ivoryos and Zenodo repository[38]. Source data are provided as a Source Data file. Source data are provided with this paper.

## Code availability

The IvoryOS source code is available in https://gitlab.com/heingroup/ivoryos and has been deposited in Zenodo repository[38].

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

## Acknowledgements

Financial support for this work was provided by the Canada Foundation for Innovation (CFI-35833, CFI-44843), Natural Sciences and Engineering Research Council of Canada (RGPIN-2021–03168, Discovery Accelerator) and the University of British Columbia to J.E.H. This research was undertaken thanks in part to funding provided to the University of Toronto's Acceleration Consortium from the Canada First Research Excellence Fund (CFREF-2022–00042).

## Author contributions

W.Z. and J.E.H. contributed to the conceptualization of the project. W.Z. and L.H. developed the software code. V.L. and R.C. contributed to the development and experimentation of the Telescope solubility platform. J.J. and J.Z. contributed to the development and experimentation of the flow chemistry platform. J.L. and Y.S. contributed to the development and experimentation of the derivatization sampling platform. M.P. and L.H. contributed to the development and experimentation of the LLE platform. M.R., W.Z., R.G., J.M. and N.D. contributed to the development of the PurPOSE platform. All authors contributed to user interface testing and suggestions. W.Z. drafted the initial manuscript and L.H., R.E., P.P. and E.T. revised the manuscript with all authors' input. J.E.H. supervised the project.

## Competing interests

The authors declare no competing interests.
