## [Transparent Peer Review file · Nature Communications]

IvoryOS: an interoperable web interface for orchestrating Python-based self-driving laboratories

Corresponding Author: Professor Jason Hein

Version 0:

Reviewer comments:

Reviewer #1

(Remarks to the Author)

The manuscript by Hein and co-workers presents a software system for closed-loop laboratory automation accessible both via a graphical user interface (UI) as well as an ergonomic Python API designed to accommodate hardware and configuration variations at point of deployment. Whilst improving UIs to minimize friction and allow precise protocol definition/execution is an important area of work, to date quite a few successful ecosystems have emerged including fully graphical UIs, software libraries accessible via standard programming languages like Python, domain specific and chemical programming languages as well as hybrid "low-code" approaches. These advances raise the bar for a significant contribution to the state-of-art that would be of interest to Nature Communications' broad readership. I am afraid in this case both the significance of results as well as the scope of methodology — very much focused on the software engineering aspect with very limited validation — does not meet the threshold for publication in my opinion. Venues with a more focused remit — RSC Digital Discovery or Journal of Open Source Software come to mind — would better represent the intended audience and level of specialization suited to this manuscript.

Regardless of the chosen venue, a much more thorough validation exercise is warranted to demonstrate the claim that the system's utility for optimization/discovery across different hardware platforms. This will need to involve execution of a variety of protocol families and deployment on at least one other hardware platform (preferably not the authors' in-house system). Without rigorous validation, the paper represents more of a software architecture manual/blueprint.

(Remarks on code availability)

The project has a helpful readme file with clear instructions to install and execute the code. Code organization seems clear enough though documentation is sparse. The frontend seems fairly straightforward to set up though without access to the specific hardware it will be a challenge to test backend functionality. The authors don't demonstrate any experimental validation so the practical definition of reproducing their work is not entirely clear.

Reviewer #2

(Remarks to the Author)

The authors introduce IvoryOS, a plug-and-play, configuration-free software designed to automatically generate snapshots of self-driving laboratories (SDLs) Python scripts. By capturing all device instances, functionalities, and argument information, IvoryOS simplifies the process of managing and documenting SDL workflows. This approach addresses a key challenge in the field: the lack of a universal, easily adaptable graphical user interface (GUI) for SDLs, which has historically limited access to advanced automation tools for many scientists. This is a significant achievement, and I want to congratulate the authors on this excellent work. I am excited to see how this promising software evolves and look forward to future updates.

Please see below a few comments/suggestions to further improve the manuscript and ensure it reaches its full potential.

1.(minor typo) Figure 3 & Figure 5: Ensure all sentences begin with capital letters.

2.(minor typo) Line 28: "And etc." is redundant; use just "etc."

3. Line 127 & 133: There are two “Step 1” entries in the plot (“Return to main panel” and “Choose device”). Consider renaming or clarifying these to avoid confusion, especially since only “Choose device” is discussed in detail.

4. Line 187-188: could you create a reference with the exact link of this script?

5. Line 193: Is there error handling implemented for scenarios where the CSV file is empty or contains many duplicate entries? It seems that it is not necessary to download an empty CSV file. I guess users should have the option to upload an existing CSV, provided it adheres to the required format.

(Remarks on code availability)

The code is well-structured, with clear README instructions and an excellent video illustration. I have the following question/comment:

In client.py, the current script appears to assume that the server is always reachable. But if the snapshot fails (e.g., due to server unavailability), the script will not generate any classes, and no meaningful error handling is in place to address this scenario.

Have you considered adding error handling for these scenarios to ensure the script handles such issues gracefully?

Version 1:

Reviewer comments:

Reviewer #1

(Remarks to the Author)

The additional of case studies featuring ivoryOS interfacing with a range of hardware platforms is a welcome addition to the manuscript and addresses the principal concern encountered when initially reviewing their contribution. A small error has been introduced in the revised abstract: "during from".

(Remarks on code availability)

Covered in my initial review — no concerns.

Reviewer #2

(Remarks to the Author)

I'd like to thank the authors for their thorough and thoughtful revisions. The updated manuscript is much clearer and more detailed, especially with the addition of the supplementary information, which really adds value to the work. I really appreciate the effort put into addressing the feedback from the previous reviews.

IvoryOS is a powerful tool for managing and documenting self-driving laboratories, and the improvements made only reinforce its potential impact. I'm genuinely looking forward to seeing how this software evolves.

I did spot a few minor typos in Figures 3 and 6 (not starting with capital letters), but these are really just small details and don't affect the overall quality of the manuscript. I recommend giving those figures another quick look to catch any remaining issues.

Congratulations once again on this great work.

(Remarks on code availability)

We acknowledge that there is a lack of thorough validation in interoperability and experimentation. We greatly appreciate the reviewers' insightful comments and have addressed them in the revised manuscript and code repository, which we are submitting for reconsideration.

To strengthen the impact of our work, we integrated ivoryOS with five previously unseen SDLs—three in-house and two externals—demonstrating its ease of integration and diverse applications. Within these new platforms, we further conducted closed-loop optimization and expanded the manuscript's discussion to highlight the advance.

In summary, our revisions emphasize how ivoryOS addresses the unique challenges of SDLs, especially in chemistry, where automation tasks are highly flexible and often require closed-loop experimentation. Currently, no available software supports such rapid adaptation, and we believe our work represents a meaningful step forward in this space.

Reviewers' comments:

Reviewer #1 (Remarks to the Author):

The manuscript by Hein and co-workers presents a software system for closed-loop laboratory automation accessible both via a graphical user interface (UI) as well as an ergonomic Python API designed to accommodate hardware and configuration variations at point of deployment. Whilst improving UIs to minimize friction and allow precise protocol definition/execution is an important area of work, to date quite a few successful ecosystems have emerged including fully graphical UIs, software libraries accessible via standard programming languages like Python, domain specific and chemical programming languages as well as hybrid "low-code" approaches. These advances raise the bar for a significant contribution to the state-of-art that would be of interest to Nature Communications' broad readership. I am afraid in this case both the significance of results as well as the scope of methodology — very much focused on the software engineering aspect with very limited validation — does not meet the threshold for publication in my opinion. Venues with a more focused remit — RSC Digital Discovery or Journal of Open Source Software come to mind — would better represent the intended audience and level of specialization suited to this manuscript.

Regardless of the chosen venue, a much more thorough validation exercise is warranted to demonstrate the claim that the system's utility for optimization/discovery across different hardware platforms. This will need to involve execution of a variety of protocol families and deployment on at least one other hardware platform (preferably not the authors' in-house system). Without rigorous validation, the paper represents more of a software architecture manual/blueprint.

Our submission was intended to participate in the SDLs software collection, aiming to provide a flexible and broadly applicable solution for streamlining the sharing and deployment of SDLs. While we acknowledge the existence of domain-specific low-code libraries and chemical

programming languages, these approaches often require substantial effort to adapt to new SDLs, as they are not designed with adaptability as a core principle. Given the increasing diversity of SDLs across the community—each with varying complexity, objectives, and frameworks—there remains a need for ready-to-use, adaptable software that facilitates rapid deployment, iteration, and collaboration without requiring continuous developer support. It is worth mentioning that this software does not conflict with further integration of other frameworks or GUIs, as there is no code modification to the current script.

We agree that demonstrating interoperability across multiple hardware platforms is essential. To strengthen our validation, we integrated ivoryOS with five previously unseen SDLs and we have incorporated two additional closed-loop demonstrations using (1) a flow chemistry system and (2) an external solubility platform. Detailed descriptions and documentation for these platforms have been added to the manuscript in Platform Integration from line 172 to line 251.

Regarding the execution of various protocol families, our software is explicitly designed to be protocol-agnostic and domain-agnostic, enabling seamless support for diverse closed-loop and high-throughput experiments. As such, validating specific protocols falls outside the intended scope of this work. However, we have restructured the manuscript to better highlight interoperability between robotic arm based and autonomous sampling-based SDLs. We have moved the software architecture discussion to the methods section to improve clarity.

Reviewer #1 (Remarks on code availability):

The project has a helpful readme file with clear instructions to install and execute the code. Code organization seems clear enough though documentation is sparse. The frontend seems fairly straightforward to set up though without access to the specific hardware it will be a challenge to test backend functionality. The authors don't demonstrate any experimental validation so the practical definition of reproducing their work is not entirely clear.

Documentation: We have significantly expanded the documentation (<https://ivoryos.readthedocs.io>), including more detailed route descriptions and additional usage examples to enhance clarity and ease of implementation.

Backend Validation: To facilitate testing and reproducibility, we have provided an abstract SDL example that allows users to validate functionality and generate sample data outputs without requiring specific hardware.

Experimental Validation: We have added physical experimentations and two closed-loop demonstrations in the manuscript, along with coding examples available in repository (/ivoryos/example/). However, reproducing experiments with specific hardware is not the primary goal of this work. For reproducibility, we recommend using the abstract SDL and provided data.

Reviewer #2 (Remarks to the Author):

The authors introduce IvoryOS, a plug-and-play, configuration-free software designed to automatically generate snapshots of self-driving laboratories (SDLs) Python scripts. By capturing all device instances, functionalities, and argument information, IvoryOS simplifies the process of managing and documenting SDL workflows. This approach addresses a key challenge in the field: the lack of a universal, easily adaptable graphical user interface (GUI) for SDLs, which has historically limited access to advanced automation tools for many scientists. This is a significant achievement, and I want to congratulate the authors on this excellent work. I am excited to see how this promising software evolves and look forward to future updates.

Please see below a few comments/suggestions to further improve the manuscript and ensure it reaches its full potential.

1.(minor typo) Figure 3 & Figure 5: Ensure all sentences begin with capital letters.

We have ensured all figures using Capital letters.

2.(minor typo) Line 28: “And etc.” is redundant; use just “etc.”

We have fixed this.

3. Line 127 & 133: There are two “Step 1” entries in the plot (“Return to main panel” and “Choose device”). Consider renaming or clarifying these to avoid confusion, especially since only “Choose device” is discussed in detail.

To improve clarity, we renamed the step number of “Return to main panel” to step 2, shifted the following step numbers accordingly.

4. Line 187-188: could you create a reference with the exact link of this script?

We have now provided a direct download link to the generated Python script in SI section 3. We have also rephrased the main text for clear SI reference. This is the generated Python script can also be found in https://gitlab.com/heingroup/ivoryos/-/blob/main/example/sdl_example/ivoryos_data/scripts/example.py

5. Line 193: Is there error handling implemented for scenarios where the CSV file is empty or contains many duplicate entries? It seems that it is not necessary to download an empty CSV file. I guess users should have the option to upload an existing CSV, provided it adheres to the required format.

If a CSV file is empty, the main experiment will not execute. If the CSV file contains duplicate entries, they are currently accepted by the script runner. However, to enhance clarity, we have added a warning message prompting users when duplicate entries are detected in the configuration input.

Reviewer #2 (Remarks on code availability):

The code is well-structured, with clear README instructions and an excellent video illustration. I have the following question/comment:

In client.py, the current script appears to assume that the server is always reachable. But if the snapshot fails (e.g., due to server unavailability), the script will not generate any classes, and no meaningful error handling is in place to address this scenario.

Have you considered adding error handling for these scenarios to ensure the script handles such issues gracefully?

We have added an initial connection check for the client before executing the script. If the snapshot reading fails due to server unavailability, the script now provides a clear error message, prompting users to check server activity or verify the URL input.

Other changes

Authors, contribution and acknowledgement: we have added more authors and clarified their contribution during the cross-platform validation and case studies.

Abstract, Introduction, and Conclusion: We have edited these sections to reflect the new integrations of IvoryOS with other SDLs.

Advance and necessity of plug-and-play software:

- In Introduction (line 72-74), we emphasized the current limitation of available solutions
- In Discussion (line 264-266, 283-286), we expanded on the protocol difference between life science and chemistry automation and highlighted the necessity of a GUI for SDL development.

Methods restructure: we moved software architecture and technical details to Methods to keep the Results section more focused.